# Non-Coding RNAs: New Biomarkers and Therapeutic Targets for Temporal Lobe Epilepsy

**DOI:** 10.3390/ijms23063063

**Published:** 2022-03-11

**Authors:** Ida Manna, Francesco Fortunato, Selene De Benedittis, Ilaria Sammarra, Gloria Bertoli, Angelo Labate, Antonio Gambardella

**Affiliations:** 1Institute of Molecular Bioimaging and Physiology (IBFM), National Research Council (CNR), Section of Germaneto, 88100 Catanzaro, Italy; 2Department of Medical and Surgical Sciences, Institute of Neurology, University “Magna Graecia”, Germaneto, 88100 Catanzaro, Italy; francescfortunato@gmail.com (F.F.); selene.db90@gmail.com (S.D.B.); ilariasammarra@gmail.com (I.S.); labate@unicz.it (A.L.); 3Institute of Molecular Bioimaging and Physiology (IBFM), National Research Council (CNR), 20090 Milan, Italy; gloria.bertoli@ibfm.cnr.it

**Keywords:** non-coding RNAs, circular RNA, long non-coding RNA, microRNA, biomarker, therapeutic target

## Abstract

Temporal lobe epilepsy (TLE) is the most common form of focal epilepsy; it is considered a network disorder associated with structural changes. Incomplete knowledge of the pathological changes in TLE complicates a therapeutic approach; indeed, 30 to 50% of patients with TLE are refractory to drug treatment. Non-coding RNAs (ncRNAs), acting as epigenetic factors, participate in the regulation of the pathophysiological processes of epilepsy and are dysregulated during epileptogenesis. Abnormal expression of ncRNA is observed in patients with epilepsy and in animal models of epilepsy. Furthermore, ncRNAs could also be used as biomarkers for the diagnosis and prognosis of treatment response in epilepsy. In summary, ncRNAs can represent important mechanisms and targets for the modulation of brain excitability and can provide information on pathomechanisms, biomarkers and novel therapies for epilepsy. In this review, we summarize the latest research advances concerning mainly molecular mechanisms, regulated by ncRNA, such as synaptic plasticity, inflammation and apoptosis, already associated with the pathogenesis of TLE. Moreover, we discuss the role of ncRNAs, such as microRNAs, long non-coding RNAs and circular RNAs, in the pathophysiology of epilepsy, highlighting their use as potential biomarkers for future therapeutic approaches.

## 1. Introduction

Temporal lobe epilepsy (TLE) is the most common type of focal epilepsy in adulthood [1]. According to seizure semiology, it can be divided into two different categories: the most common mesial TLE (mTLE), in which seizures originate from mesial and internal structures of the temporal lobe, and a rarer form with lateral temporal lobe symptoms [2]. Anti-seizure medications (ASMs) are the first-line therapy in patients with mTLE, but resistance to ASMs is a major clinical challenge in its treatment [3]. Furthermore, although mTLE has been traditionally viewed as an acquired drug-resistance epilepsy, for which surgical treatment is often considered, a milder drug-sensitive form has also been recognized [2,4]. Thus, there is an urgent need to better understand the molecular mechanisms underlying mTLE and develop new molecular-based therapeutic approaches.

Our understanding of mTLE derives primarily from clinical, imaging and physiological data from humans and animal models. Further, mTLE can be caused by various pathological processes, the most common of which is hippocampal sclerosis (HS), as demonstrated by post-resection and post-mortem studies [5]. The etiology of HS, which is in vivo defined by detecting hippocampal atrophy on brain magnetic resonance imaging, is multifactorial, with combined hereditary/environment influences [6]. Moreover, HS can be related to post-natal hippocampal vulnerability to the injuries resulting from prolonged febrile seizures or head trauma. Experimental studies revealed that neuronal damage and gliosis in HS cause alterations in neuronal connectivity and enhance excitability, which contributes to epileptogenesis [7].

The aberrant network restructuring and hyperexcitability have been associated with large scale changes in gene transcription and protein expression. Several genome-wide studies also support evidence that epilepsy progression is accompanied by large scale epigenetic modifications in TLE. The pathological mechanism underlying TLE, although not yet fully understood, could involve aberrant regulation of gene expression, including post-transcriptional networks. In transcriptional process regulation and the control of epigenetic gene expression, the non-coding RNA (ncRNA) molecules play an important role. Recent developments have indicated that a significant part of the genome is actively transcribed as ncRNA molecules [8]. The ncRNAs are emerging as key actors in the regulation of biological processes and act as a class of functional RNAs that regulate gene expression in a post-transcriptional manner [9]. Further, ncRNAs are a class of RNA molecules that do not encode proteins; they are highly expressed in the brain and are involved in the regulation of physiological and pathophysiological processes, including neuronal development, apoptosis, neurogenesis, oxidative stress, synaptic plasticity and immune system activation, processes implicated in mTLE [10,11,12]. Moreover, ncRNAs may be grouped into different classes and classified according to size and function: microRNAs (miRNAs), long non-coding RNAs (lncRNAs) and circular RNAs (circRNAs). Further, miRNAs are small ncRNAs, about 20 to 23 nucleotides, which regulate gene expression by binding to the 3′-untranslated region of target RNAs; this binding can lead to mRNA degradation or translation inhibition and thus regulates the expression of target genes [13]. Additionally, miRNAs act as negative regulators: the up-regulation of miRNAs may down-regulate their target mRNA; therefore, an altered miRNA expression can result in dysregulation of key genes and pathways that contribute to disease development [14].

Compared to miRNAs, lncRNAs, longer than 200 nucleotides, remain poorly understood regarding their expression and roles as they can regulate gene expression through complex molecular mechanisms. It is known that lncRNAs can be transcribed with different orientations than the coding genes: some are transcribed from regions that overlap multiple exons of another coding transcript (sense lncRNA), others overlap with antisense strand coding genes (antisense lncRNA) and other lncRNAs derive from non-coding DNA sequences, such as introns, or from elements regulators, such as enhancers. Furthermore, some of them are transcribed from intergenic regions and have their own promoters and regulatory elements [13]. Growing evidence defines lncRNAs as regulators of several biological processes, and their expression seems to be tightly regulated both in physiological conditions and in various diseases [15]. Further, circRNAs are a new type of ncRNA that differs from linear RNAs in that a covalent bond connects the 3′ end of a downstream exon to the 5′ end of an upstream exon, forming a closed circular structure. This latter structure and the lack of 3′ or 5′-cap polyadenylates make them resistant to exonuclease compared to linear RNA [13]. Moreover, circRNAs can affect the gene expression at the transcriptional or post-transcriptional level by acting as miRNA sponges, i.e., circRNAs sequester miRNAs and repress their endogenous activity, preventing these molecules from binding to their messenger RNA targets [16]. Therefore, it was proposed that circRNAs can control neuronal mechanisms by regulating, through miRNA sponging, the degradation or translation mRNA targets [17]. The mechanisms of action of ncRNAs and biological functions are summarized in Figure 1.

Experimental studies have demonstrated alterations in post-transcriptional regulation in mTLE, and alteration of the expression and function of various classes of ncRNAs has been highlighted both in experimental models of TLE and in patients with TLE. All these observations suggest that ncRNAs can participate in the epileptogenesis process [18,19,20,21]. Numerous studies have highlighted the possible relationship between ncRNAs and the pathological mechanisms underlying TLE. 

Further studies on the mechanisms of action of ncRNAs will improve our understanding of the processes of epileptogenesis and will lead, in the near future, to new methods of diagnosis and treatment. In this review, we consider the main aspects of ncRNAs by summarizing the current knowledge on the role of the latter in the pathogenesis of TLE, highlighting the possibility of using ncRNAs as disease biomarkers by discussing their potential translational impact as therapeutic targets.

## 2. Role of miRNAs in TLE Pathogenesis

### 2.1. Studies Suggested the Role of miRNA in TLE

It has been estimated that around 60% of human proteins could be directly regulated by miRNAs by binding to complementary sites on mRNAs and decreasing mRNA stability and translation [22]. It is well known that mTLE is associated with dysregulated hippocampal gene expression, and that this altered gene expression is due, in part, to the miRNAs. The latter are short ncRNAs that control protein levels by binding target mRNAs, and their abnormal expression is thought to be related to pathways of inflammation, cell death, synaptic reorganization and neuronal excitability, processes strongly associated with epileptogenesis. In recent years, accumulating evidence suggested that miRNA was implicated in the pathogenesis of epilepsy [23]. Most studies have focused on the function of brain-specific miRNAs in the mouse model or on samples from TLE patients (see Korotkov et al. for detailed review) [24]. Nudelman et al., using the pilocarpine mouse model of epilepsy, reported that neuronal activity rapidly induces transcription of the CREB-regulated miR-132 in vivo, and this was the first study to demonstrate a change in miRNA after seizures [25]. In a chronic TLE rat model, Song et al. showed a dysregulated profile of miRNAs. In particular, by monitoring the expression of two miRNAs, miR-let-7e and miR-23a /b, they observed dynamic change expression of these miRNAs over time after status epilepticus [26]. Bencurova et al. performed miRNA profiling on the largest cohort of patients with mTLE+HS so far. They observed altered miRNAs expression in hippocampal tissue samples of patients undergoing surgery for pharmacologically refractory TLE. In particular they identified an altered expression of 20 microRNAs in mTLE with hippocampal sclerosis (HS), of which 19 miRNAs were up-regulated and one down-regulated. The uniqueness of this study is in the complete evaluation of the whole human hippocampal miRNome using NGS to identify miRNAs potentially participating in mTLE+HS pathology. This study, building on growing evidence indicating that miRNAs contribute to the pathophysiology of seizures and epilepsy, extends the current knowledge of miRNA-mediated gene expression regulation in mTLE+HS by identifying miRNAs with altered expression in mTLE+HS [27]. In a recent work, in order to more accurately predict the regulatory potential of a given miRNA than measuring the overall miRNA levels in a sample, Veno et al. performed small RNA sequencing (RNA-seq) of Ago2-loaded miRNAs to define functionally engaged microRNAs in the hippocampus of three different animal models. Using a multimodel sequencing approach, they were able to demonstrate that there are shared miRNAs dysregulated at all the phases in the development of epilepsy, up to and including the period of active chronic epilepsy. Most of the miRNA changes fell within a 1.5- to 3-fold range, although some, including miR-142a-5p, displayed much larger fold changes. Their results identify shared, dysregulated and functionally active microRNAs during the pathogenesis of epilepsy, which could represent therapeutic antiseizure targets [18].

### 2.2. Principal Pathogenic Mechanisms and Related miRNAs Involved in TLE

To date, the etiology of epilepsy is controversial, although its pathogenesis may be related to neuronal cell apoptosis and inflammatory response [28]. Several functional studies suggest that miRNAs may be involved in the occurrence and development of epilepsy by affecting these pathological processes. Much evidence underlies the involvement of miRNAs in the inflammatory and immune process in TLE, and, about this, their role has been extensively studied [29]. Alteration in miRNA expression may be involved in epilepsy pathogenesis by regulating the expression of inflammatory factors, such as IL-1, INF-α and TNF-α. Of note, an increased expression of miR-146a was observed in reactive astrocytes in hippocampal sclerosis specimens of TLE patients [30]. In the refractory TLE rat model, miR-146a increased the epilepsy susceptibility by reducing complement factor H. Hence, epilepsy-induced reduction of miR-146a differential expression could decrease the occurrence of epilepsy [31]. Another associated miRNA with the regulation of the inflammatory pathways in epilepsy is miR-155. Ashhab et al. first investigated the dynamic expression patterns of TNF-α and miR-155 in the hippocampi of a rat model and children with mTLE. TNF-α and miR-155, having similar expression patterns in the three stages of mTLE development, and their relationship at the astrocyte level, may suggest a direct interactive relationship during mTLE development. The increased TNF-α levels operate as a feedback loop to regulate miR-155 expression; therefore, these two molecules interact to mediate the inflammatory process. They showed that the expression levels of miR-155 and TNF-α were increased in children with chronic TLE, and miR-155 could increase TNF-α expression and, consequently, the inflammatory response [32]. Moreover, miRNAs are not only involved in inflammatory processes but several research studies highlight their contribution in the regulation of mechanisms such as apoptosis. First, Tivnan and colleagues demonstrated an association between miR-34 and apoptosis by decreasing MAP3K9 mRNA and protein levels [33]. Subsequently, Hu et al. observed an increase in pro-apoptotic miR-34a expression in the post-status epilepticus rat hippocampus. Further, miR-34a was significantly up-regulated at 1 day, 7 days and 2 weeks post-status epilepticus and at 2 months after temporal lobe epilepsy. Experiments with the miR-34a antagomir revealed that targeting miR-34a led to an inhibition of activated caspase-3 protein expression, which may contribute to increased neuronal survival and reduced neuronal death or apoptosis. Moreover, miR-21 expression in the hippocampus is increased many hours after a seizure, thereby reducing the inhibitory effect on the 3′ UTR region of the neurotrophin-3 and promoting cell apoptosis [34]. The activation of pro-apoptotic genes that promote neuronal apoptosis may be at the basis of the mechanism of action of miR-21 in epilepsy [35]. Lu et al., in models of epilepsy, induced using kainic acid, found, in the hippocampus of epileptic rats, a significant increase in miR-27a-3p expression. A miR-27a-3p inhibitor also inhibited apoptosis of hippocampal neurons in epileptic rats, promoted Bcl2 expression and reduced Bax and Caspase3 expression [36]. Fan et al. used a rat model of epilepsy and epilepsy-induced hippocampal neurons to study the function of miR-15a. The results of the present study suggested that the expression levels of miR-15a were down-regulated in TLE tissues and epileptic cells. The authors also discovered that the up-regulated expression levels of miR-15a significantly inhibited the rate of apoptosis in epileptic cells [37]. Another study showed that the expression of miR-135b-5p was significantly decreased in children with TLE and in the epileptic rat neuron model. The miR-135b-5p increased expression attenuates the postepileptic influence on cell viability and apoptosis by targeting SIRT1 [38]. All these data suggest the contribution of several miRNAs to the pathogenesis of epilepsy by affecting apoptosis, suggesting that modulation of these dysregulated miRNAs could provide novel strategies for the treatment of epilepsy.

Several subsets of miRNA have been proposed as a potential regulator of a variety of processes involved in epilepsy, such as neuronal function and synaptic plasticity. For instance, Bot et al. decided to investigate changes in the expression levels of miRNAs in the dentate gyrus, a structure that has been studied for a long time in the context of epilepsy [39]. Indeed, in the epileptic dentate gyrus, the occurrence of abnormal neuronal plasticity and abnormal neurogenesis limited to certain populations of neurons have been observed [40]. Bot and colleagues described the miRNA expression patterns at different times following epileptogenic stimulus in the animal model of temporal lobe epilepsy. They showed for the first time that prolonged changes in the expression of miRNAs in the rat dentate gyrus follow status epilepticus, and they presented lists of miRNAs that change expression levels [40]. Further, miR-134 is the first miRNA involved in remodeling of neuronal structures following epilepsy [41]. Moreover, this miRNA is also involved in the control of the neuronal microstructure and regulates the size of dendritic spines [42]. Kaalund et al. investigated miRNAs expression using microarray, qRT-PCR and in situ hybridization, detecting miR-218 and miR-204 down-regulation in human hippocampal specimens of MTLE patients with HS. Moreover, they validated miR-218 mediate repression of ROBO1, GRM1, SLC1A2 and GNAI2 genes involved in axonal guidance and/or synaptic plasticity [43]. A mir-132 up-regulation was detected in hippocampal tissues obtained from children with MTLE [35]. The intraperitoneally injection of ant-132 in a pilocarpine model mouse can reduce chronic recurrent seizures and seizure frequency; moreover, an inhibition of aberrant dendritic spine formation was observed with mir-132 silencing in the hippocampus of mice. These effects could be related to the interaction of mir-132 with the pathway involved in the regulation of morphology and electrophysiology of dendritic spines [44]. In conclusion, future studies, including assessing the impact of miRNAs on the proteome, are required to understand the complex alterations of neuronal function and network defects in epilepsy.

## 3. Impact on Diagnosis and Prognosis

### 3.1. miRNAs as Putative Biomarkers in TLE

Besides their role as modulators of cellular activity, in pathological and non-pathological conditions, miRNAs are also released from the cells into biological fluids. Following their release, miRNAs can circulate in different forms, such as in complex with Argonaut 2 (AGO2) protein or encapsulated in microvesicles, including the exosomes. This provides stability in an extracellular environment, where they are protected from endogenous RNase activity; moreover, even at low levels, they can be detected with fast and reliable methods, such as microarray, qRT-PCR or sequencing [45]. Finally, alteration of miRNAs signature in biofluids correlates with pathophysiological conditions, pointing out that circulating miRNAs could be potential biomarkers reflecting the state of the organism. All these features make these molecules particularly suitable for use as biomarkers and have been extensively studied for CNS disorders. Thus, miRNAs have been proposed as biomarkers for brain injury and neurodegenerative diseases such as Parkinson’s disease [46] and Alzheimer’s disease [47]. Several studies have evaluated the potential of miRNAs as biomarkers related to epilepsy, including TLE [48]. Avansini and colleagues carried out a high-throughput sequencing analysis on plasma of 14 patients with mTLE, 13 patients with focal cortical dysplasia (FCD) and 16 healthy subjects. They observed that miR-134 was significantly down-regulated in mTLE patients, highlighting that low hsa-miR-134 expression might be a potential non-invasive biomarker to support the diagnosis of mTLE [49]. In a subsequent study aimed at identifying robust blood-based biomarkers for TLE diagnosis, Raoof et al. identified a set of three miRNAs with a diagnostic biomarker potential: miR-27a-3p, miR-328-3p, miR-654-3p. The authors employed two different genome-wide screening platforms and plasma samples collected from patients coming from two different countries of origin in order to overcome bias related to platform-specific technology and a single clinical cohort [50]. In the blood of mTLE patients with hippocampal sclerosis (mTLE-HS), four other potential circulating biomarkers: miR-145, miR-181c, miR-199a and miR-1183, were found to be overexpressed [51]. Another important serum biomarker for the diagnosis of mTLE-HS is miR-328-3p, with high area under the curve (AUC) values when comparing controls to Engel I (90.3%). As a peripheral biomarker to predict the surgical prognosis of patients with mTLE-HS, miR-654- 3p had the statistical power (AUC = 73.6%) to differentiate Engel I from Engel III-IV patients [52]. Circulating miRNAs have been associated with drug-resistant epilepsy. In this regard, Wang et al. analyzed the differential expression of serum miRNAs in 30 drug-resistant and 30 drug-sensitive patients by Illumina HiSeq2000 sequencing technology. They found that miR-301a-3p is the biomarker that best discriminates between drug-resistant epilepsy and drug-sensitive epilepsy. Furthermore, multiple regression analysis highlighted that down-regulated miR-301a-3p expression represents a potential biomarker for the diagnosis (sensitivity of 81.5% and specificity of 81.2%) [53]. In a more recent study, Leontariti et al. found that mir-146a and mir-134 correlated with the risk of developing drug resistance [54]. Brennan et al. (2020) performed the first multi-model, genome-wide profiling of plasma miRNAs during epileptogenesis and in chronic temporal lobe epilepsy animals, and they identified a set of dysregulated microRNAs. Of note, in validation studies, they found that miR-93-5p, miR-199a-3p and miR-574-3p were also dysregulated in plasma from patients with intractable temporal lobe epilepsy [55]. The current studies identify additional circulating miRNA biomarkers of experimental and human epilepsy, which may support diagnosis of TLE. Since the discovery of predictive biomarkers and early identification of pharmacoresistant patients is of the highest priority, we have recently investigated the serum expression of some miRNAs previously related to epilepsy and/or to drug resistance mechanisms. Further, qRT-PCR analysis revealed a significant up-regulation of miR-142-5p, miR-146a-5p and miR-223-3p in TLE patients compared to healthy controls. More importantly, our work brings out the potential of miR-142 and miR-223 as predictive biomarkers in pharmacological response. The results suggest that they could be good biomarkers to classify drug-sensitive vs. drug-resistant epileptic patients, with an AUC of 0.80 for miR-142-5p and 0.75 for miR-223-3p [56]. Despite the promising results, the cohort size is small and validating the results in a larger study court will be needed. Comprehensive findings regarding miRNAs as potential biomarkers in drug-resistant epilepsy, and their involvement in the development of resistance mechanisms, were exposed in a recent review by Bohosova et al. [57]. In conclusion, all these data show how epilepsy can be associated with changes in the expression of certain circulating miRNAs. However, adding more studies performed with coherent detection techniques might increase the potential for using miRNAs as biomarkers for epilepsy. The recent functional and biomarker studies need to be replicated by other groups to build a more robust evidence base. Table 1 summarizes the dysregulated miRNAs that have been detected in different biological fluids of patients with TLE.

### 3.2. Prospects for miRNAs Therapeutics in TLE

There are no doubts that microRNAs play a key role in many biological processes, so it is not surprising that dysregulated expression of microRNAs is observed in all diseases, including epilepsy. Thus, microRNA research allows not only to identify molecular and cellular pathways that contribute to epilepsy and epileptogenesis but also allows the development of tools for therapy. However, due to the multi-targeting and multi-pathway actions of miRNAs, prediction and therapeutic function are not easy to approach. Several functional studies have shown the potential therapeutic targets of epilepsy on miRNA levels through administration of chemically engineered antisense oligonucleotides that target specific miRNAs called mimics or antagomirs, which are capable of reproducing or inhibiting specific miRNA function, respectively [58]. It emerged from data that prolonged seizures in rodents cause up-regulation of miR-134, and levels of miR-134 are also higher in the brain of patients with drug-resistant TLE. Furthermore, miR-134 is enriched in dendrites of hippocampal neurons, where it negatively regulates spine volume [59]. Jimenez-Mateoset et al., since targeting miR-134 in vivo using antagomirs had potent anticonvulsant effects against kainic acid-induced seizures and was associated with a reduction in dendritic spine number, measured dendritic spine volume in mice injected with miR-134-targeting antagomirs and tested effects of the antagomirs on status epilepticus triggered by the cholinergic agonist pilocarpine. This study provides in vivo evidence that pre-administration of miR-134 antagomirs regulates spine volume in the hippocampus and attenuates the seizure degree of epileptic mice, indicating broad conservation of anticonvulsant effects [60]. Yuan et al. observed a reduction in the level of electrical excitability in epileptic neurons due to the silencing of miR-132. Furthermore, silencing of miR-132 inhibits the abnormal development of dendritic spines and reduces chronic spontaneous seizures in lithium–pilocarpine-induced epileptic murine models. [44]. In a previous work, Jimenez-Mateos et al. showed that neuronal death can be associated with up-regulation of miR-132 and pretreatment with miR-132 antagomirs may reduce hippocampus injuries after seizures [61]. These results demonstrate the neuroprotective effect of miR-132 silencing and that miR-132 may serve as a putative target for the development of antiepileptic treatment. Recently, an attempt has been made to control epilepsy in mouse models through the regulation of miR-146a expression. In this regard, Tao and colleagues note that intranasal administration of miR-146a, in the acute phase of lithium–pilocarpine-induced epilepsy, could ameliorate both the onset of epilepsy and hippocampal injury through modulating the expression of inflammatory factors [62]. Wang X. et al. found that intracerebroventricular injection of miR-146a can reduce seizures in a rat model of lithium–pilocarpine-induced status epilepticus [63]. To conclude, miR-146a can regulate inflammatory factors involved in the onset of epilepsy and be both a biomarker for diagnosing epilepsy and an important therapeutic target. However, although these results refer to few animal model studies and have rarely been verified in vivo, they demonstrate that more miRNAs are potential therapeutic targets for the treatment of disease. In order for them to be used as diagnostic tests that could support patient treatment and prognosis, they should be extensively studied in different epilepsy models and validated in humans

## 4. Role of lncRNAs in TLE Pathogenesis

### 4.1. Studies Suggested the Role of lncRNA in TLE

To date, the contribution of lncRNAs in the pathogenesis of TLE is shown by studies in experimental animal models of disease. Among these works, the first study was conducted by Lee et al. to explore the role of lncRNAs by comparing their expression in pilocarpine and kainite models and control mice to investigate epileptic mechanisms. They identified 384 and 279 altered lncRNAs in the pilocarpine model and in the kainate model, respectively. These dysregulated lncRNAs could affect the occurrence and development of epilepsy through different mechanisms [64]. In this regard, Yang et al. explored the role of lncRNAs and mRNA in the pilocarpine mouse model in specific brain regions, the hippocampus and cortex, which are the most important structures in the epileptogenesis of TLE, and compared the results to those of the control mouse. The authors found a total of 22 and 83 LncRNAs were up- and down-regulated (≥2.0-fold, all *p* < 0.05), respectively, in the hippocampus of the epilepsy model, while 46 and 659 lncRNAs were up- and down-regulated, respectively, in the cortex of the epilepsy model. Gene ontology (GO) and pathway analysis reported that the dysregulated mRNAs were closely related to the well-known biological processes underlying epileptogenesis. Furthermore, as TLE is known to damage the hippocampus, resulting in sprouting of mossy fibers, gliosis and atrophy, analysis of the protein–protein interaction showed that altered protein-encoding transcripts were linked with the mammalian of rapamycin (mTOR) and silencing transcription factor RE-1 (REST) pathways, which are involved in cell signal transduction and epigenetic mechanisms, respectively, during the HS process [65]. In summary, this is the first study to analyze the profiles of dysregulated lncRNAs in the pilocarpine mouse model that might provide additional evidence of the mechanisms of epileptogenesis and a therapeutic choice for TLE. In a recent work *Cui* et al. analyzed the expression profiles of lncRNAs in human mTLE with HS. They used microarray analysis to analyze the differential expression of lncRNAs and mRNAs in three HS and three control hippocampus samples. Their results showed that 497 lncRNAs were differentially expressed in hippocampal sclerosis samples compared to control hippocampus, i.e., 294 lncRNAs were up-regulated and 203 down-regulated. Furthermore, 399 differentially expressed mRNAs were identified, with 236 up-regulated and 163 down-regulated. The GO and pathway analyses revealed the 30 most significantly enriched terms, which were related to inflammation and neuropeptide receptor activity, that are predicted to play roles in mTLE; in particular, lncRNA RP11-414J4 may contribute to epileptogenesis by targeting CPLX3 [66]. This study is the first to profile the hippocampal expression of lncRNAs in patients with HS and to compare this with that in the normal hippocampus. Despite these encouraging results, the study presents some limitations: a) the number of samples in each group was low, making it necessary to validate in a larger cohort of patients of different ethnicities; b) therapy with several ASMs may have caused an altered expression of lncRNAs; c) the samples in the control group were derived from post mortem tissues; therefore, delays in autopsy may have altered the lncRNA profiles.

### 4.2. Principal Pathogenic Mechanisms and Related lncRNAs Involved in TLE

It is well known that proinflammatory cytokine expression in glia is significantly increased in various experimental seizure models and in human epilepsies [67], and this suggests that the inflammatory response plays a very important role in the occurrence and development of epilepsy. In recent years, it emerged that lncRNA does not only regulate the gene expression of the physiological process and signal pathway in the nervous system but also regulates the expression of pro-inflammatory and anti-inflammatory cytokines [68]. Wan et al. highlighted that lncRNA NEAT1(nuclear-enriched abundant transcript 1) influences the inflammatory response. In particular, they found the expression level of NEAT1 was markedly increased in the hippocampal neurons of epilepsy patients. Furthermore, they investigated the role of NEAT1 in epilepsy, as well as examined the function of the miR-129-5p and Notch signaling pathways in an epilepsy model in vitro. They also revealed the regulation mechanism of the NEAT1, miR-129-5p and Notch signaling pathways in the occurrence of epilepsy, thus providing a potentially viable new direction and target for epilepsy clinical treatment [69]. In another work, Zhang et al. found that epilepsy modeling increased the expression of IL-1β, IL-6 and TNF-α in the hippocampus, and that overexpression of lncMEG3 reduced the expression of IL-1β, IL-6 and TNF-α in the hippocampus of rats with TLE. These findings indicate that lncMEG3 had the function of inhibiting inflammation in rats with TLE [70]. An important cause of neuronal damage due to the onset and state of epilepsy is apoptosis, and alterations in apoptosis-associated signaling pathways have been widely reported in TLE tissue and animal TLE models [71]. A notable role as modulators of the apoptotic response is given by Bcl-2 family protein and caspase, including pro-apoptotic (Bax) and anti-apoptotic members (Bcl-2). Regarding this, reduced levels of Bcl-2 and enhanced levels of Bax cleaved caspase-3 were detected in apoptotic neurons in the hippocampus of a KA-induced epilepsy model [72]. Growing data have highlighted a role of lncRNAas in regulating apoptosis. For example, Han et al. used a genome-wide approach in order to reveal the functions and regulatory mechanisms of lncH19, the first identified lncRNA. They identified many genes that were differentially expressed when H19 was overexpressed or knocked down in a KA-induced epileptic rat model [73]. In another work, again, Han et al. found that lncH19 was the most differentially expressed lncRNA in the hippocampus of epileptic rat models. Of note, it was down-regulated in the acute period of convulsions and, on the contrary, up-regulated in the seizure-free period of TLE. The authors speculated that, in the chronic period, repeated and spontaneous seizures induce hypoxia-reoxygenation, resulting in altered levels of H19 and, ultimately, hippocampal damage. Moreover, they also point out that, in the hippocampus of the TLE rat model, H19 stimulates neuron apoptosis by operating as competing endogenous RNA (ceRNA) to sponge miRNA let-7b in the regulation of Casp3 expression. Finally, they point out that an overexpression of H19 induced the activation of microglia and astrocytes through modulation of the JAK/STAT pathway and the stimulation of release of proinflammatory cytokines in the hippocampus, as already suggested previously [74,75]. Wang and collaborators suggested that lncRNA-UCA1 is capable of inducing or worsening epilepsy in a pilocarpine-induced rat model by interacting with NF-kB and that UCA1 expression in peripheral blood correlated positively with that of brain tissue. Indeed, they observed higher expression levels of both genes in epileptic rats than in the control group, and a higher level of UCA1 mRNA was also observed in peripheral blood of the study group compared to the control group. These results highlight the involvement of lncUCA1 in the pathogenesis of the disease [76]. In a more recent study, Wang and collaborators demonstrated that lncRNA-UCA1 suppressed hippocampal astrocyte activation and JAK/STAT/GLAST expression in the rat models of temporal lobe epilepsy and improved the adverse reactions caused by epilepsy. Thus, they speculated that the overexpression of UCA1 might be involved in the TLE progression via inhibiting astrocyte activation and GLAST expression in hippocampi of epileptic rats [77].

Many data highlighted that several lncRNAs are actively implicated in the regulation of synaptic plasticity [78]. BDNF antisense RNA (BDNF-AS) is a lncRNA that is transcribed from the opposite strand of brain-derived neurotrophic factor (BDNF), a neurotrophin that is engaged in neuronal differentiation, synaptic plasticity and memory processes. With regard to epilepsy, a study showed that the expression of BDNF is increased in human neocortex removed as a treatment of intractable seizures, while the levels of BDNF-AS are significantly reduced [79]. These results indicate that the mRNA–lncRNA interplay may represent a regulatory network of human brain plasticity and that inhibiting the BDNF pathway could be used as a potential therapeutic strategy for epilepsy. MALAT1, which is also known as nuclear-enriched abundant transcript 2 (NEAT2), is the most studied lncRNA. This lncRNA is ubiquitously expressed, is very abundant in the brain and regulates several genes that are interested in dendritic and synapse development [80]. Regarding its role in epilepsy, recently, Wu et al. found that, in rats with epilepsy, MALAT1 depletion mediated by anti-MALAT1 siRNA resulted in activation of the PI3K/Akt signaling pathway and loss of hippocampal neurons. LY294002, an inhibitor of the PI3K/Akt signaling pathway, could reverse the events caused by MALAT1 knockdown. Taken together, these findings indicate that down-regulation of MALAT1 activates the PI3K/Akt signaling pathway to protect hippocampal neurons against autophagy and apoptosis in rats with epilepsy [81].

## 5. Impact on Diagnosis and Prognosis

### 5.1. lncRNA as Putative Biomarkers in TLE

To date, the function of lncRNAs has been explored both in experimental models of animal epilepsy and in human tissue. lncRNAs within the brain exhibit specific temporal and spatial patterns [64]. Several studies have shown that hundreds of lnRNAs are differentially expressed; furthermore, it has been pointed out that dysregulated lncRNAs with co-dysregulated mRNAs may be potential clinical targets for the epigenetic modulation of chronic epilepsy [65]. As a consequence of the complex genetics of epilepsy, there is a considerable need for well-defined differential diagnostic criteria, which might allow for specific treatment. There are currently no well-defined biomarkers for epilepsy that aid in the diagnosis, which is primarily clinical. Circulating lncRNAs show particular characteristics to act as biomarkers, even if the mechanism related to their extracellular releases is still not fully understood. lncRNAs can be considered a new class of non-invasive diagnostic and prognostic biomarkers due to their abundance and long-term stability in body fluids, such as blood, serum or plasma, and easy accessibility, especially if included in apoptotic bodies or exosomes [82]. The study of Hashemian et al. considers four specific lncRNAs in the peripheral blood of patients compared to controls (N = 40 per group): HOXA-AS2, SPRY4-IT1, MEG3 and LINC-ROR. In particular, lncRNAs, HOXA-AS2 and SPRY4-IT1 showed greater expression in the comparison between the patient and control groups [83]. Another study used human peripheral blood from TLE patients and compared the lncRNA ILF3-AS1 levels between serum and hippocampal tissues. It is known that ILF3-AS1 potentiates inflammatory cytokines and TNF-alpha expression together with increased matrix metalloproteinases, all of which are associated with epilepsy. In this study, the authors highlighted that the ILF3-AS1 levels in the hippocampal and serum are higher in TLE patients compared to healthy control subjects, suggesting lncRNAs utility to monitor the development of TLE [84]. In a recent work, conducted on blood, the expression of seven lncRNAs (UCA1, MALAT1, NKILA ANRIL, DILC, PACER, THRIL) was evaluated in 80 epileptic patients (40 refractory and 40 non-refractory ones) and 40 normal individuals. Significant differences in the expression of all the assessed lncRNAs except for DILC and MALAT1 were detected between the study groups. Finally, the authors demonstrated outstanding diagnostic power for a number of lncRNAs, especially in the differentiation of non-refractory patients from controls [85]. Taken together, these studies demonstrate, in the peripheral blood of epileptic patients, dysregulation of lncRNAs and classify them as enhanced biomarkers for this neurological condition.

### 5.2. Prospects for lncRNAs Therapeutics in TLE

The possibility of a targeted use of lncRNA to prevent or delay the onset of epilepsy is promising, even if it is still in the early stages. lncRNAs are targetable using an antisense-based oligonucleotide (ASOs) inhibitors approach [86]. lncRNAs, transcribed in the antisense direction to coding genes negatively affect the latter in cis. A newsworthy evolution is the investigation of natural antisense transcripts (NATs). Indeed, ASOs that target NATs, termed ‘antagoNATs’, have demonstrated notable preclinical results for gene reactivation in the central nervous system. For example, antagoNATs effectively increase brain-derived neurotrophic factor (BDNF), a protein highly involved in memory formation [87]. To date, there is only one study that used oligonucleotide inhibitors of a NAT lncRNA to up-regulate expression of the SCN1A mRNA; the human SCN1A gene locus expresses a NAT that functions to limit SCN1A proteins levels in cells. The introduction of an oligonucleotide targeting the SCN1A-NAT compounds (AntagoNATs) reported a resulting specific up-regulation of SCN1A, both in vitro and in vivo, in the brain of the Dravet knock-in mouse model and in a non-human primate. AntagoNAT-mediated up-regulation of SCN1A in postnatal Dravet mice led to significant improvements in seizure phenotype and excitability of hippocampal interneurons [88]. In conclusion, though, the role of lncRNAs in epileptogenesis is beginning to be explored, and lncRNA therapeutic-based strategies likely open a new scenario for future therapeutic opportunities for epilepsy.

## 6. Role of circRNAs in TLE Pathogenesis

Recently, circRNAs emerged as a new class of endogenous ncRNAs that could regulate gene expression at the transcriptional or post-transcriptional level, working as miRNA sponges, as they are capable to bind miRNAs, inhibiting the binding between miRNAs and their messenger RNA (mRNA) targets [16,89]. Hence, an indirect neuronal control mechanism for circRNAs has been proposed; i.e., through interactions with disease-related miRNA, they can control the degradation or translation mRNA targets [17]. The understanding and results in studying circRNAs are still very limited at the moment, and the biological functions of circRNAs in TLE are unclear. However, the regulatory role of circRNAs in the brain takes place in neurotransmission events, synaptic plasticity, apoptosis and other aspects of neural activity [90]. This indicates that circRNAs can participate in the control of synaptic function and neuroplasticity, processes with an important role in the development of epilepsy. Future research will increase our understanding of the regulation and function of these circRNAs.

### 6.1. Studies Suggested the Role of circRNA in TLE

Recent studies have shown that dysregulated circRNAs might have a pathophysiologic role in chronic epilepsy by regulating multiple disease-relevant mRNAs via circRNA–miRNA–mRNA interactions. To date, investigations into the importance of circRNAs in epilepsy have been based on chronic stages of TLE, both in patient samples and in a mouse model. In the work of Gong et al., to look for specific circRNAs differentially expressed between drug-resistant TLE patients and control samples of temporal neocortical tissues, the authors subjected five samples from each group to circRNA microarray assays. They found 586 circRNAs differently expressed between the TLE groups and the control tissues. Furthermore, they observed a significant reduction in the expression of circRNA-0067835 in tissues of 22 surgical patients with TLE compared with controls, and, subsequently, they examined the potential miRNAs associated with circRNA-0067835. They showed that circRNA-0067835 was reduced in the tissues and plasma of 22 surgical TLE patients, while miR-155 expression was increased in TLE, and, furthermore, decreased expression of circRNA-0067835 correlated to increased seizure frequency in TLE patients. In conclusion, their results showed that circRNA-0067835 is an important modifier of the pathogenesis of pharmacoresistant TLE by acting as a miRNA sponge for miR-155 [20]. Lee et al. hypothesized that circRNAs might be correlated with the pathophysiology of chronic epilepsy, and, in this regard, in a model of murine epilepsy, they evaluated the altered circRNA expressions and their possible regulatory effects on their target miRNAs and mRNAs. The authors evaluated the profile of differentially regulated circRNAs in the hippocampus of a pilocarpine epilepsy model and tried to identify their functions in the epi-genetic regulatory processes involved in the pathophysiology of chronic epilepsy. In the hippocampus of the pilocarpine model, the overall expression profile of circRNAs highlighted 43 circRNAs: 26 up-regulated and 17 down-regulated. Furthermore, as circRNAs contain miRNA binding sites, so-called miRNA response elements (MREs), that enable circRNAs to sequestrate the target miRNA, a process known as the “miRNA sponge effect”, they might modulate the expression of their target genes via circRNA–miRNA–mRNA regulatory networks. In support of this, the authors found that the change in the expression of MREs in dysregulated circRNAs had a negative association with their target miRNA expression [91]. However, future and further analyses should investigate the possible regulatory network among dysregulated circRNA, miRNA and mRNA and their pathophysiologic role in chronic epilepsy. Recently, Gray et al. studied circRNA expression in the temporal cortex and hippocampus of patients with drug-resistant mTLE and healthy subjects, and they identified aberrant expression of circRNA in mTLE patients. They also hypothesized that dysregulated circRNA expression disrupts gene networks in mTLE by sequestering miRNAs from their target genes. Their work, firstly, demonstrated changes within the temporal cortex and hippocampus of mTLE patients, in particular one differentially expressed circRNA in the cortex and eight in the hippocampus of mTLE patients, including circRNA-HOMER1, which is expressed in synapses. Of note, seven of these nine differentially expressed circRNAs exhibited miRNA sponging features and appeared to have phenotypic roles in mTLE by contributing to HS [21]. Despite these results, further experimental confirmations are needed to explain the mechanisms of circRNA–miRNA binding. In a more recent work, Gomes-Duarte et al., to explore specific circRNA–miRNA–mRNA network deregulation in the early stages of experimental epilepsy, used a perforant pathway stimulation (PPS) rat model of TLE and performed RNA-sequencing on hippocampal tissue of the rat model of TLE. PPS is characterized by elevated survival rates and reduced variability of hippocampal injury during hippocampal atrophy, gliosis and specific neuronal death of the hippocampal region, which are the major pathophysiological features of MTLE. The PPS animal model was utilized to profile circRNA expression at different stages during the process of epileptogenesis. This analysis revealed 218 differentially expressed circRNA; the authors also observed that most of these circRNAs were modified at the time of the first spontaneous seizure. Of note, their studies demonstrated that circ_Arhgap4 and circ_Nav3 are predicted to interact with miR-6328 and miR-10b-3p, respectively, and could indirectly regulate expression of miR-6328 and miR-10b-3p targets. For the first time, they examined the regulation of circRNAs in the course of epilepsy development [92].

### 6.2. Principal Pathogenic Mechanisms and Related circRNAs Involved in TLE

It is well known that circRNAs are highly expressed in neurons, particularly during synaptogenesis, but the mechanism of action of most circRNAs remains unexplored. The hippocampus neural network contains a large number of neurons, and its dysfunction plays a fundamental role in TLE [93]. Although the vital role of circRNAs in epilepsy is recognized, few studies have been conducted on circRNAs in in vivo or in vitro models of epilepsy. In a recent study, Chen et al. considered the circ_0003170, which was highly expressed in TLE specimens. In their research, an epilepsy cell model was established by the treatment of Mg^2+^-free in human neurons-hippocampal (Hn-h) cells. The expression of circ_0003170 was monitored in TLE serum specimens and Mg^2+^-free- induced neurons, and the function of circ_0003170 on cell viability, cycle, apoptosis and oxidative stress was investigated in vitro. Their results showed that circ_0003170 was aberrantly up-regulated in TLE serum specimens and cell models. Finally, the authors noted that, in Mg^2+^-free-induced Hn-h, knockdown of circ_0003170 ameliorated Mg^2+^- free-induced cell cycle arrest, cell apoptosis and oxidative stress of human hippocampal neurons, thus protecting the normal functions of hippocampal neurons [94]. Another study involved the circUBQLN1 and HN-h cell model of epilepsy. CircUBQLN1 was down-regulated in epilepsy samples and an Mg^2+^-free-induced cell model. Functional analysis in vitro suggested that circUBQLN1 overexpression facilitated proliferation but reduced apoptosis and oxidative stress in Mg^2+^-free-treated HN-h cells. The circUBQLN1 expression was significantly inhibited in 23 TLE tissues and Mg^2+^-free-treated HN-h cells compared with eight normal tissues and the untreated HN-h cells. The abnormal expression of circUBQLN1 was confirmed in TLE. In conclusion, these findings demonstrated that circUBQLN1 overexpression attenuated the Mg^2+^-free-induced neurological damages, including proliferation repression, apoptosis promotion and oxidative response, in HN-h cells [95]. Future studies, which will be based on these first results, will help us to determine in a more precise way new mechanisms that regulate the progression of TLE through inflammatory signaling pathways.

It is well known that circRNAs are abundant in brain tissue and have specific distribution; in fact, they are mainly expressed in neutropils, in particular in dendrites, and participate in synaptic function regulation and neuroplasticity. This suggests, consequently, that circRNAs might participate in epileptogenesis. The available data suggest that circRNAs have important roles in synaptic plasticity and neuronal function (see *Hanan* et al. for detailed review) [96]. It has been shown that synaptic reorganization of mossy fiber projections of the TLE hippocampus has an important function in persistent cellular hyperexcitability in the epileptic circuitry of the hippocampal formation [97]. Li et al. found that 442 circRNAs were dysregulated in the TLE cortex, and, subsequently, they selected 10 mostly dysregulated circRNAs for validation. In particular, they chose the most up-regulated (circ-EFCAB2) and down-regulated circRNAs (circa-DROSHA). These circRNAs arise from exons and could regulate their host genes: up-regulation of EFCAB2 could influence calcium ion binding and, therefore, may concur to epilepsy; the loss of DROSHA causes neuronal and glial abnormality and seizures. These results show that circ-EFCAB2 and circ-DROSHA could be involved in the process of TLE by regulating EFCAB2 and DROSHA [98]. While this may underlie novel TLE mechanisms, the confirmation of these circRNA–miRNA-target gene axes needs to be further investigated in TLE models. In a recent interesting experiment, Zheng et al. established that increased circ-DROSHA weakened the neural injury of the TLE cell model. To study the mechanism of circ-DROSHA, they discovered, using bioinformatics analysis, that circ-DROSHA could bind to miR-106b-5p to mediate the expression of myocyte-specific enhancement factor 2C (MEF2C); that is, circ_DROSHA regulated MEF2C expression via sponging miR-106b-5p. In conclusion, circ-DROSHA up-regulation alleviated the cytotoxicity of the TLE cell model by enhancing cell proliferation and repressing cell apoptosis [99]. All these results obtained so far demonstrate the involvement of circRNAs in epileptogenesis, but further studies are needed to confirm these results. Table 2 summarizes some of the main pathogenetic mechanisms of TLE that can be regulated by ncRNAs.

## 7. Impact on Diagnosis and Prognosis

### circRNAs as Therapeutic Targets and Diagnostic Biomarkers in TLE

It is the case that circRNAs are more stable than linear RNAs due to their greater stability, and this feature offers a huge clinical advantage for their use as diagnostic and therapeutic biomarkers for TLE. Thanks to the development of sequencing technology, many circRNAs were found in TLE tissue. A first study that identified the global expression profiles and characteristics of circRNAs in TLE human temporal cortex tissue was conducted by Li et al. They collected human temporal cortex tissue from 17 TLE patients and 17 non-TLE patients, and they discovered that a large number of circRNAs were imbalanced in TLE tissues through high-throughput data. In particular, 442 circRNAs were dysregulated among the TLE and non-TLE groups; of these circRNAs, 188 were up-regulated and 254 were down-regulated in the TLE patient group. Of note, circ-EFCAB2 was overexpressed, while circ-DROSHA expression was significantly reduced in the TLE group compared to the non-TLE group. Bioinformatic analysis predicted that these circRNAs can work as a “miRNA sponge” through regulating genes in the predicted circRNA–miRNA gene axis. In this regard, the authors find that interactions of circEFCAB2 with miR-485-5p and circ-DROSHA with miR-1252-5p were highly correlated with the expression of epilepsy-associated genes CLCN6 and ATP1A2, respectively [98]. In the work of Gong et al., in agreement with brain tissue data, they highlighted a significant reduction in circRNA-0067835 expression in the plasma of 22 surgical TLE patients compared to controls. Further, to better know the potential role of this circRNA-0067835 in the progression of TLE, the authors evaluated the association with clinical features of patients with TLE. They observed that lower circRNA-0067835 correlated to increased seizure frequency and HS. Furthermore, the authors showed that circRNA-0067835 is a significant modifier of the pathogenesis of drug-resistant TLE by acting as a miRNA sponge for miR-155 and thereby promoting the function of the FOXO3a signaling pathway [20]. Collectively, these results suggested that circRNA-0067835 could have a key role in the progression of TLE. Zheng et al., to determine the involvement of circ-DROSHA in TLE pathogenesis, measured its expression level in the serum samples of 35 TLE patients and 35 non-TLE subjects. They found that circ_DROSHA expression was prominently reduced in the serum of TLE patients compared with the control group. Although the study relates to a small cohort, these findings suggest that circulating circ-DROSHA might be a promising biomarker for the clinical diagnosis of TLE [100]. Future studies are needed to expand the circ-DROSHA analysis in TLE to a larger cohort of patients and demonstrate that reduced circ-DROSHA expression is related to disease. In summary, these studies show that circRNA dysregulations may reflect the pathogenesis of TLE and could represent potential therapeutic targets and biomarkers in TLE patients. A summary of the expression of longRNAs and circRNAs in brain and blood samples of TLE patients is presented in Table 3.

## 8. Summary: Outlooks and Next Steps

Temporal lobe epilepsy is commonly drug-resistant and is associated with dysregulated hippocampal gene expression. Although AED are usually successful in controlling seizures, 30 to 40% of TLE patients have seizures despite receiving adequate medical management, and those with pharmacoresistant TLE often require surgical treatment, which often causes memory and cognitive impairments. Indeed, there is no effective treatment to prevent the development of epilepsy, so the identification of biomarkers could greatly help patient management. Currently, a reliable biomarker could help optimize drug treatment and block or slow further progression of the disease. Furthermore, in the search for new biomarkers, it is essential to consider accessibility in clinical use; therefore, they should be easily identified with common clinical laboratory equipment. In the case of TLE, the identification of biomarkers could lead to the development and use of antiepileptic drugs directed against the underlying cause of epilepsy or even before patients develop recurrent seizures. For this reason, the molecular mechanisms underlying epileptogenesis, including epigenetic mechanisms, are, therefore, key focal points in the study and identification of epileptic biomarkers because they can change with the onset and the progression of the disorder, although there are still important gaps in our current understanding regarding ncRNAs and many of their functions remain to be understood more deeply and clarified. Literature data prove that ncRNAs are dysregulated in experimental and human epilepsy. Since epigenetic markers are also detectable in circulating biofluids, and thanks to their abundance and stability in circulating fluids, they can be considered diagnostic biomarkers. In this regard, miRNAs have been studied the most due to their specific tissue expression, stability and ease of analysis [101]. Moreover, miRNAs represent a new class of signaling molecules able to cross the gap junction channels (GJs), protein complexes located in the plasma membrane that allow direct communication between adjacent cells. This crossing represents a general mechanism of intercellular gene regulation and a promising approach for miRNA delivery [100]. The hypothesis that GJs are involved in the mechanisms underlying seizures is well known, and evidence has shown that disruption of intercellular communication through astrocyte gap junctions is a key event in epileptogenesis. In the recent work by Volnova et al., it is highlighted that specific regulators of GJs activity in astrocytes have great potential in antiepileptic therapy [102]. Recently, van Vliet et al. have proposed a protocol for the standardization of procedures for the discovery of circulating microRNA biomarkers in rat models of epileptogenesis, with the aim to facilitate clinical biomarker discovery for epileptogenesis in man by preclinical standardization. This aspect is fundamental because, once standard criteria for the identification of biomarkers are established and applied, the studies conducted in the different laboratories will become more comparable [103]. Ultimately, this will lead to new, easily evaluable diagnostic tests and will provide more insight into the disease mechanisms, enabling the development of new therapies to treat epilepsy in patients at risk. Other epigenetic markers may also have diagnostic value, such as as lncRNAs and circRNAs, and different searches have been carried out in clinical patients and different animal models, demonstrating dysregulated profiles of these epigenetic markers. Therapeutic targeting of ncRNAs constitutes a promising and valid approach for the treatment of diseases. Various RNA-based therapies, as well as antisense oligonucleotides, anti-microRNA, miRNA mimics, miRNA sponges and therapeutic circular RNA, have been developed. Animal experiments already indicate some promising targets for the use of ncRNAs as therapeutic tools in epilepsy as the use of miR antagonists for miR-134 or mimic-miRs for miR-22 was capable of reducing the neuronal death and seizure severity in animal models, although these and other examples of preclinical uses of miRNAs for the treatment need further study. Hsiao et al., using oligonucleotide-based compounds (AntagoNATs) targeting SCN1A-NAT (CUR-1916), were able to induce specific up-regulation of SCN1A, both in vitro and in vivo, in the brain of a Dravet knock-in mouse model and a non-human primate. CUR-1916 is the first antisense oligonucleotide targeting a non-coding RNA to reach clinical trials for epilepsy. However, the main problem with all antisense oligonucleotides is delivery. They cannot pass from the circulation to the brain as they are too large to cross the blood–brain barrier. They must then be injected intrathecally [88]. Recent advances in antisense technology have led to the approval of an antisense oligonucleotide for the treatment of spinal muscular atrophy. The oligonucleotide is administered intrathecally through the lumbar puncture, thus providing a more targeted delivery to the CNS [104]. This approach, although still invasive, could be used for miRNA-based therapies in epilepsy when therapeutic intervention fails or intervention surgery is not indicated. In conclusion, ncRNAs are emerging as key players in the pathology of TLE: as they are involved in many molecular mechanisms underlying the disease, they can be used as biomarkers for better diagnosis and/or assessment of disease progression. Finally, ncRNAs are also promising targets for new therapeutic strategies to be employed in the treatment of epilepsy.

## Figures and Tables

**Figure 1 ijms-23-03063-f001:**
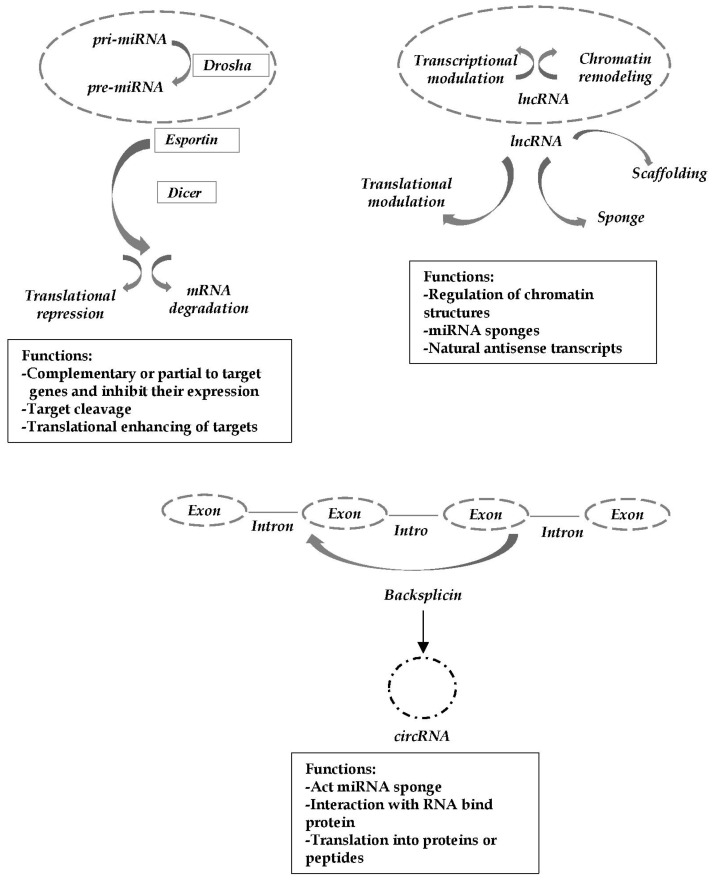
Schematic representation of ncRNA mechanisms of action. ncRNAs regulate gene expression both at transcriptional and post-transcriptional levels. miRNA, lncRNA and circRNA act in a variety of ways, promoting or inhibiting the expression of specific targets. Mechanisms of action of ncRNAs are summarized within each box.

**Table 1 ijms-23-03063-t001:** Summary of studies on miRNA profiling in TLE.

Significantly Dysregulated miRNAs	Source	Cohort Composition	Methods	Regulation	*p*-Value, AUC	Reference
miR-134	plasma	14 TLE 16 HC 65 TLE * 83 HC *	qRT-PCR	**↓** **↓**	0.018, 75% 0.0003, 67%	[49]
miR-27a-3p miR-328-3p miR-654-3p miR-654-3p miR-27a-3p miR-328-3p	plasma	64 refractory focal epilepsy 32 HC 102 TLE * vs. 110 HC *	OpenArray qRT-PCR	**↑** **↑** **↑** **↑** **↑** **↑**	<0.05, 63% >0.05 0.05, 63% 0.05, 63%	[50]
miR-145 miR-181c miR-199a miR-1183	blood	10 mTLE-HS 20 HC	qRT-PCR	**↑** **↑** **↑** **↑**	0.005 0.03 0.01 0.001	[51]
miR-328-3p	serum	28 TLE 11 HC	qRT-PCR	**↑**	<0.001, 93%	[52]
3 miRNAs 12 miRNAs miR-301a-3p miR-194-5p miR-30b-5p miR-342-5p miR-4446-3p	serum serum	30 DRE vs 30 DSE 77 DRE vs. 45 HC 77 DRE vs. 81 DSE	NGS qRT-PCR	**↑** **↓** **↓** **↓** **↓** **↓** **↓** **↓**	<0.05 <0.0001, 89% <0.0001, 74% <0.0001, 68% <0.033, 72% <0.004, 70% <0.0001	[53]
miR-146a-5p miR-134	serum	86 DRE vs. 76 DSE	qRT-PCR	**↑** **↑**	0.002, 64% 0.01, 61%	[54]
miR-93-5p miR-199a-3p miR-574-3p	plasma	20 intractable TLE 16 HC	qRT-PCR	**↑** **↑** **↑**	<0.05, 77% <0.05, 80% <0.05, 79%	[55]
miR-142-5p miR-146a-5p miR-223-3p miR-142-5p miR-223-3p	serum	27 TLE vs. 20 HC 10 DRE vs. 17 DSE	qRT-PCR qRT-PCR	**↑** **↑** **↑** **↑** **↑**	0.001 0.02 < 0.001 <0.001, 80% <0.001, 75%	[56]

TLE: temporal lobe epilepsy; DRE: drug-resistant epilepsy; DSE: drug-sensitive epilepsy; HC: healthy controls; vs: versus; qRT-PCR: quantitative reverse transcription PCR; NGS: next generation sequencing; AUC: area under the curve; * cohort of validation; ↑ or ↓symbolize up- or down-regulation of miRNAs expression.

**Table 2 ijms-23-03063-t002:** Principal pathogenic mechanisms and related ncRNAs involved in TLE.

Pathaways	miRNAs	lncRNAs	circRNAs
Neuroinflammation	miR-146a [30,31]	lncNEAT1 [69]	
	miR-155 [32]	lncMEG3 [70]	
		lncH19 [75]	
		lncUCA1 [76,77]	
Apoptosis/Neuronal loss	miR-34a [33,34]	lncH19 [74]	circ_0003170 [94]
	miR-21 [35]	lncMALAT1 [81]	circ_UBQLN1 [95]
	miR-27a-3p [36]		circ_DROSHA [99]
	miR-15a [37]		
	miR-135b-5p [38]		
Synaptic plasticity	miR-134 [41,42]	lncBDNF-AS [79]	
	miR-218 [43]	lncMALAT1 [80]	
	miR-132 [44]		

**Table 3 ijms-23-03063-t003:** Summary of studies on lncRNAs and circRNAs profiling in TLE.

Significantly Dysregulated lncRNAs	Source	Cohort Composition	Methods	Regulation	*p*-Value	Referenece
HOXA-AS2 SPRY4-IT1	blood	40 TLE vs. 40 controls	qRT-PCR	↑ ↑	0.001 0.02	[83]
ILF3-AS1	blood temporal cortex (lesionectomy) control tissues (intracranial hematoma)	23 TLE vs. 18 controls	qRT-PCR	↑ ↑	0.001 0.001	[84]
UCA1 NKILA ANRIL THRIL	blood	40 DRE vs. 40 HC 40 DSE vs. 40 HC	qRT-PCR	↑ ↑ ↑ ↓	0.003 <0.0001 0.018 0.006 <0.0001 0.019 0.04 <0.0001	[85]
**Significantly dysregulated circRNA**						
circ-EFCAB2 circ-DROSHA	temporal cortices	17 TLE 17 HC	qRT-PCR	↑ ↓	<0.05 <0.05	[98]
circRNA-0067835	temporal cortex (lobectomy) control tissues (temporal neocortical) plasma	5 TLE 5 HC 22 TLE 22 HC	Microarray qRT-PCR	↓ ↓	<0.01 <0.01	[20]

TLE: temporal lobe epilepsy; vs.: versus; HC: healthy controls; DRE: drug-resistant epilepsy; DSE: drug-sensitive epilepsy; qRT-PCR: quantitative reverse transcription PCR; ↑ or ↓ symbolize up- or down-regulation of ncRNAs expression.

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
