# Peer review of "Non-Coding RNAs: New Biomarkers and Therapeutic Targets for Temporal Lobe Epilepsy"

_ijms, 2022, doi:10.3390/ijms23063063_

Round 1

Reviewer 1 Report

The review paper summarizes an interesting and relevant topic. There have not been similar previous papers dealing with very similar content so writing on ncRNA in temporal lobe epilepsy was imminent. The paper is well structured and appropriate references are used. There are however, some points, which can be further improved.

  1. Letters in figure 1 are small and sometimes distorted. This should be corrected.
  2. Additional figure(s) and table(s) would help to focus the paper.
  3. The results of high-throughput techniques are often presented. The authors of these papers use very different criteria for up/down regulation. It would be helpful for the reader to know how strict the criteria were in a given article (e.g. p<0.05 or q<0.05, or FDR was calculated). It would also be very important to know if a “change” was actually validated with independent technique in the reviewed paper for the particular RNA.
  4. It is often not described what tissue was used for the reviewed measurement of RNA. For biomarker chapters, it is often plasma (or sometimes CSF?). For tissue measurement, it would be always good to know if it is brain tissue at all, which brain region was dissected, whether is it surgical or post-mortem sample. Were single cell/nucleus transcriptomics applied?

Minor points:

  1. ASM is written as AMS in some cases.
  2. Letter types vary within the text.
  3. Lateral temporal lobe epilepsy is introduced as a separate type in the Introduction but this line is not followed at all later on.
  4. Hs is abbreviated sometimes as HS
  5. “This structure and lack polyadenylated 3’-tail or 5’-cap, making them exonuclease resistant than linear RNA [13].” sentence should be grammatically corrected.
  6. “Ago2” was used but not properly introduced.
  7. “This last are short ncRNAs which control….” sentence should be grammatically corrected.
  8. “Antagomir” was used but not properly introduced.
  9. “It is well known that the expression of proinflammatory cytokines in glia of seizures is significantly increased…” sentence should be grammatically corrected.

Author Response

Comments to Reviewer 1:

We thank the reviewer for the positive comments.

  1. Letters in figure 1 are small and sometimes distorted. This should be corrected.

Answer: We regret for the mistake in the Figure 1. We have increased the font size and paid attention to its correct formatting.

  1. Additional figure(s) and table(s) would help to focus the paper.

Answer: Following the reviewer’s suggestion, we have now added the table that contains principal ncRNAs involved in the pathogenic mechanisms underlying TLE.

  1. The results of high-throughput techniques are often presented. The authors of these papers use very different criteria for up/down regulation. It would be helpful for the reader to know how strict the criteria were in a given article (e.g. p<0.05 or q<0.05, or FDR was calculated). It would also be very important to know if a “change” was actually validated with independent technique in the reviewed paper for the particular RNA.
  2. It is often not described what tissue was used for the reviewed measurement of RNA. For biomarker chapters, it is often plasma (or sometimes CSF?). For tissue measurement, it would be always good to know if it is brain tissue at all, which brain region was dissected, whether is it surgical or post-mortem sample. Were single cell/nucleus transcriptomics applied?

Answer: Following the reviewer’s criticism, we have added 2 tables that summarize the results obtained in the different studies included in our manuscript.

Minor points

  1. ASM is written as AMS in some cases.
  2. Letter types vary within the text.
  3. Lateral temporal lobe epilepsy is introduced as a separate type in the Introduction but this line is not followed at all later on.
  4. Hs is abbreviated sometimes as HS
  5. “This structure and lack polyadenylated 3’-tail or 5’-cap, making them exonuclease resistant than linear RNA [13].” sentence should be grammatically corrected.
  6. “Ago2” was used but not properly introduced.
  7. “This last are short ncRNAs which control….” sentence should be grammatically corrected.
  8. “Antagomir” was used but not properly introduced.
  9. “It is well known that the expression of proinflammatory cytokines in glia of seizures is significantly increased…” sentence should be grammatically corrected.

Answer: we have corrected the typos. Regarding the query on lateral temporal lobe epilepsy, in the introduction we have reported the international classification of TLE syndromes. The present manuscript focuses on mesial TLE which is the most common form of focal epilepsy in adulthood.

Reviewer 2 Report

The manuscript “Non-coding RNAs: new biomarkers and therapeutic targets for Temporal Lobe Epilepsy” by Manna et al are reviewing the interesting methodology employing the use of non-coding RNAs (which are dysregulated during epileptogenesis) as biomarkers for the diagnosis and prognosis of treatment response in temporal lobe epilepsy (TLE). Authors discuss the role of specific ncRNAs, such as microRNAs (miRNAs), long non-coding RNAs (lncRNAs), and circular RNAs (circRNAs), in the pathophysiology of epilepsy, highlighting their use as potential biomarkers for future therapeutic approaches.

It is a new and fast developing arena of research and may be interesting to many researchers in the field. The article is  generally well written in good, at least  understandable English.  Unfortunately, there are some parts that need improvement:

  1. Some phrases are heavy and not adequately edited. For example:

Page 4. Parr 2.1 Phrase :

In a recent work, Veno et al. to more accurately predict the regulatory potential of a given miRNA than measuring the overall miRNA levels in a sample, they performed small RNA sequencing (RNA-seq) of Ago2-loaded miRNAs to define functionally engaged microRNAs in the hippocampus of three different animal models.

While this phrase is understandable, Reviewer suggest changes, to made it more “English”. I can suggest:

In a recent work, in order to more accurately predict the regulatory potential of a given miRNA than measuring the overall miRNA levels in a sample, Veno et al. performed small RNA sequencing (RNA-seq) of Ago2-loaded miRNAs to define functionally engaged microRNAs in the hippocampus of three different animal models.

  1. On Page 14 SUMMARY

The first paragraphs of this summary are using different letter sizes then in the main part of the text, Please correct.

  1. Figure 1 Have multiple problems of line shifts (text in boxes and near the arrows), probably because the figure was wrongly inserted in text. Please, correct

  1. This Reviewer also suggest that authors add at least a phrase about transport of miRNAs through gap junctions (contacts between cells), especially in the view of importance of gap junctions between brain cells in epilepsy ( see references):

Lemcke H, Steinhoff G, David R. Gap junctional shuttling of miRNA--A novel pathway of intercellular gene regulation and its prospects in clinical application. Cell Signal. 2015 Dec;27(12):2506-14. doi: 10.1016/j.cellsig.2015.09.012.

Volnova A, Tsytsarev V, Ganina O, Vélez-Crespo GE, Alves JM, Ignashchenkova A, Inyushin M. The Anti-Epileptic Effects of Carbenoxolone In Vitro and In Vivo. Int J Mol Sci. 2022 Jan 8;23(2):663. doi: 10.3390/ijms23020663.

After these small corrections the article may  be accepted for publication

Author Response

Comments to Reviewer 2:

Thank you very much for the interest in our manuscript and for all your suggestions.

  1. Some phrases are heavy and not adequately edited
  2. On Page 14 SUMMARY
  3. Figure 1 Have multiple problems of line shifts (text in boxes and near the arrows), probably because the figure was wrongly inserted in text. Please, correct
  4. This Reviewer also suggest that authors add at least a phrase about transport of miRNAs through gap junctions (contacts between cells), especially in the view of importance of gap junctions between brain cells in epilepsy ( see references):

Answer: we have made all the corrections and added the references suggested by the reviewer.

Round 2

Reviewer 1 Report

The authors responded to the critiques very well. Consequently, he manuscript improved a lot.